# The Potential Role of Fecal Microbiota Transplant in the Reversal or Stabilization of Multiple Sclerosis Symptoms: A Literature Review on Efficacy and Safety

**DOI:** 10.3390/microorganisms11122840

**Published:** 2023-11-22

**Authors:** Tooba Laeeq, Tahne Vongsavath, Kyaw Min Tun, Annie S. Hong

**Affiliations:** 1Department of Internal Medicine, University of Nevada, Las Vegas, NV 89154, USA; 2Department of Gastroenterology, University of Nevada, Las Vegas, NV 89154, USA

**Keywords:** multiple sclerosis, autoimmunity, fecal microbiota transplant, demyelination, intestinal barrier, microbial diversity, short-chain fatty acids, Vitamin K

## Abstract

Multiple sclerosis (MS) affects millions of people worldwide, and recent data have identified the potential role of the gut microbiome in inducing autoimmunity in MS patients. To investigate the potential of fecal microbiota transplant (FMT) as a treatment option for MS, we conducted a comprehensive literature search (PubMed/Medline, Embase, Web of Science, Scopus, and Cochrane) and identified five studies that involved 15 adult MS patients who received FMT for gastrointestinal symptoms. The primary outcome of this review was to assess the effect of FMT in reversing and improving motor symptoms in MS patients, while the secondary outcome was to evaluate the safety of FMT in this patient population. Our findings suggest that all 15 patients who received FMT experienced improved and reversed neurological symptoms secondary to MS. This improvement was sustained even in follow-up years, with no adverse effects observed. These results indicate that FMT may hold promise as a treatment option for MS, although further research is necessary to confirm these findings.

## 1. Introduction

Multiple sclerosis is the most prevalent chronic, immune-mediated, inflammatory disease involving the central nervous system (CNS), that is, the brain and spinal cord, and is known to affect 2.8 million people worldwide as estimated by surveys conducted in 2020 to determine disease incidence [1]. These surveys reached 84% of the countries that reported the prevalence of multiple sclerosis in 2020 as compared with 71% in 2013, with an alarming increase in multiple sclerosis across the globe [1]. An alarming feature is the highest disease burden in the younger population, as it is one of the most common non-traumatic disabilities among this age group [2]. The prevalence of multiple sclerosis varies with the highest rates in Europe and North America, with the age of onset peaking between 20 and 40 years [3].

Perturbation in the intestinal barrier plays a huge role in disease pathology by controlling systemic inflammation [4]. Intestinal barrier breakdown is associated with central nervous system demyelination, with microbial derivatives entering stomach circulation and impacting microglial functions [4]. Risk factors for multiple sclerosis also include smoking tobacco, the Epstein–Barr virus, recurrent infections, obesity, low vitamin D levels, and reduced exposure to sunlight, among others [5]. Interestingly, protective factors include oral tobacco use, increased coffee consumption, and evidence of seropositivity for cytomegalovirus infection in the serum [5]. The literature suggests anti-inflammatory properties are associated with CMV infection, leading to immune evasion in MS patients and minimizing the inflammatory response [6]. Epstein–Barr virus has long been associated with lymphomas, suggesting key roles in multiple sclerosis via the modulation of the immune system [7]. Another study revealed a 32-fold increase in multiple sclerosis after Epstein–Barr virus infection with higher levels of neurofilament light chain, which is a biomarker of neuroaxonal degeneration after EBV serum conversion [8]. The leading reason that multiple sclerosis is common in temperate regions is also decreased sunlight, causing vitamin D deficiency, which plays a crucial role in the differentiation of oligodendrocytes from progenitor cells [9,10]. Smoking and obesity are thought to interfere with human leukocyte antigens.

Genes that alter the adaptive immunity pathway lead to multiple sclerosis [11]. Multiple sclerosis results from complex interactions between genetic and environmental factors [5,12]. 

Despite recent advances, the diagnosis of multiple sclerosis remains clinical [13]. Clinical cues include dissemination in space and time, excluding alternative diagnoses for multiple sclerosis [13]. In this setting, McDonald’s criteria were designed in 2001 to prevent the misdiagnosis of multiple sclerosis, including MRI, neurological history, and laboratory data [14]. However, these criteria have required multiple revisions, with the most recent being in 2017 [14]. To avoid misdiagnosis, it is important to use the appropriate criteria when diagnosing typical MS-related demyelination after carefully excluding other possible diagnoses. These criteria also help to reduce the misdiagnosis of juxtacortical and periventricular lesions on MRI by clearly defining them as lesions that abut the ventricles and the cortex. They have redefined the size threshold for MS lesions, unlike previous criteria, as at least 3 mm in the long axis, also aiding differentiation from other disease processes. Furthermore, the inclusion of positive CSF-restricted oligoclonal bands in disseminated time criteria has increased the specificity and positive predictive value of early MS diagnosis. However, the use of these criteria falls short when evaluating atypical MS patients [15]. 

The hallmark of multiple sclerosis is the presence of demyelinating lesions in the central nervous system, characterized by inflammatory infiltration and the breakdown of the protective blood–brain barrier [16]. Multiple sclerosis has various presenting symptoms, which can involve the motor nervous system, the sensory nervous system, visual pathways, and the brainstem [17]. The first clinical event for most patients is usually optic neuritis, incomplete myelitis, or brainstem syndrome; demyelinating lesions on MRI are the most important predictor for recurrence in patients with clinically isolated syndromes [17]. Different types of multiple sclerosis include Relapsing–Remitting (RRMS), Secondary Progressive (SPMS), Primary Progressive (PPMS), and Progressive Relapsing (PRMS), and the defining features of these stages are included in Table 1. 

Unfortunately, the decreasing age of onset in recent years has warranted aggressive and effective treatment strategies to prevent the lifelong morbidity and mortality associated with multiple sclerosis [19]. Despite recent advances in the treatment of multiple sclerosis, there sadly remains no cure [19]. Treatments for multiple sclerosis are divided into the management of acute relapse versus disease-modifying treatments versus symptomatic management [17]. An acute relapse, once confirmed with MRI, is treated with a high dose of corticosteroids, which shorten the duration of the relapse [20]. As an adjunct to high-dose corticosteroids, plasma therapy can also be used in severe cases for acute management [20]. Disease-modifying therapy modulates the function of T lymphocytes and B lymphocytes in the disease pathology [21]. The goal of disease-modifying therapy is the prevention of long-term morbidity and disability [21]. Disease-modifying therapies include interferon beta, glatiramer acetate, natalizumab, and fingolimod, which have variable responses in patients with multiple sclerosis [21]. Despite these interventions, symptom management remains key in patients with multiple sclerosis; it reduces functional disabilities and improves the quality of life in patients affected by multiple sclerosis [22]. This review discusses fecal microbial transplant as a potential treatment modality for multiple sclerosis-related symptoms.

Recent studies have shown an improvement in the symptoms of multiple sclerosis with fecal microbial transplant, which is intriguing and fascinating [23]. Bidirectional communication between our gut and central nervous systems has established a pathway that can be modulated for the treatment of different autoimmune diseases [23]. This crosstalk—which is affected by a plethora of things, including our environment, drugs, dietary factors, and genetics, among others—is known as the microbiome–gut–brain axis [24]. This captivating flow includes the immune, circulatory, and neural pathways, providing new targets for treatment [25]. Despite this groundbreaking observation, data are scarce that support this argument. The purpose of this first-of-its-kind review is to highlight the existing data on fecal microbiota transplant as a treatment option for multiple sclerosis. 

## 2. Methods

### 2.1. Search Strategy

We performed a comprehensive literature search across five databases (PubMed/Medline, Embase, Web of Science, Scopus, and Cochrane) using variations of the keywords “fecal microbiota transplant” and “multiple sclerosis” to identify original studies published from inception through 30 June 2022. Results were limited to human studies published in English. There was a total of 755 studies for review. See Appendix A for detailed search terms.

### 2.2. Eligibility Criteria

Inclusion criteria: (1) Patients with multiple sclerosis and baseline motor symptoms with or without GI symptoms; (2) fecal microbiota transplant as treatment; (3) reporting of patient data and outcomes after first fecal infusion; (4) patients of any sex; (5) minimum follow-up time (3 weeks); and (6) studies of all levels of quality of evidence. 

Exclusion criteria: (1) Studies without patient data; (2) non-English studies; (3) animal studies; and (4) patients with other neuromotor disorders.

### 2.3. Study Outcomes

Primary study outcomes for this study included the stabilization and reversal of the neurological symptoms of multiple sclerosis patients secondary to fecal microbiota transplant. Neurological symptoms were assessed using the Expanded Disability Status Scale (EDSS) (n = 3), the Multiple Sclerosis Functional Composite Score (MSFC) (n = 1), and the Multiple Sclerosis Walking Scale (MSWS-12) (n = 1). The EDSS remains the most widely used tool for multiple sclerosis and is reliable and effective based on the evaluation of functional systems with an EDSS score of 0, which is defined as normal neurological function [26]. The MSFC is a multidimensional clinical outcome measure consisting of three key dimensions: leg function and ambulation, arm and hand function, and cognitive function [27]. Lastly, the MSWS-12 measures the disease impact of multiple sclerosis on walking abilities by assessing the walking speed, endurance, and gait quality in multiple sclerosis patients [28].

Our secondary outcome included an assessment of the safety of fecal microbiota transplant in multiple sclerosis patients. All patients included in the study were assessed and followed up for the efficacy and possible adverse effects of fecal microbiota transplant, including a worsening of multiple sclerosis, new symptoms, allergic reactions, and anaphylactic reactions. 

### 2.4. Study Selection and Data Extraction

A total of 755 articles were retrieved on the initial search (PubMed: 510, Embase: 184, Cochrane 141). Two authors (T.L. and T.V) independently reviewed these titles and abstracts, after which, 438 (PubMed: 377, Embase: 61, Cochrane: none) articles were deemed relevant to patient data. Full texts were then reviewed by at least two of the following authors, T.V and T.V, after which, 5 remaining studies fulfilled complete eligibility criteria. In cases of disagreement, a senior reviewer (A.S.H.) arbitrated the final decision for inclusion. A summary of the included studies is shown in Table 2. IRB review was not required as all data were extracted from the published literature, and no patient intervention was directly performed.

## 3. Results

In our review, all included subjects who received fecal microbiota transplant for their gastrointestinal symptoms had improvement and reversal regarding their neurological symptoms secondary to multiple sclerosis. Gait abnormalities are some of the most prominent disabilities associated with multiple sclerosis [29]. A prominent effect was seen in the gait/ambulation of included patients with multiple sclerosis who underwent fecal microbiota transplant as evident in Table 3, Table 4, Table 5, Table 6, Table 7 and Table 8. Six patients noticed an improvement in ambulation and strength and a reduction in weakness the in lower extremities. Moreover, Engen’s study showed a significant and sustained increase in brain-derived neurotrophic factor (BDNF) with improved gait metrics, as illustrated in Table 3 [30]. Overall, the impact of fecal microbiota transplant on gait metrics was substantial, significantly reducing morbidity and dysfunction secondary to multiple sclerosis. 

Our review also depicted the increased safety of fecal microbiota transplant in multiple sclerosis patients, as out of the 15 patients included in this study, no lethal/deadly adverse effect was seen and no serious adverse effect was reported either. Kait’s study did report a mild adverse reaction of hives in one patient directly correlated to fecal microbiota transplant [29]. 

## 4. Discussion

The astounding discovery of the potential role of gut dysbiosis in the pathophysiology of multiple sclerosis has opened doors for the improved management of multiple sclerosis. In this review, we discuss the efficacy of fecal microbiota transplant in improving neurological symptoms among patients with multiple sclerosis. In our study, all included subjects who received fecal microbiota transplant for their GI symptoms had a secondary improvement and, in some cases, reversal regarding their neurological symptoms secondary to multiple sclerosis. Even in follow-up years (ranging from 1 year to 14 years, as indicated in Table 2) after receiving fecal microbiota transplant, many participants remained in remission [32]. It was also found that previously wheelchair-bound patients were able to walk unassisted after treatment, demonstrating impressive improvement in motor function and mobility [32]. The selective sensitivity of the blood–brain barrier plays a crucial role in demyelination among multiple sclerosis patients [34]. T and B lymphocytes play a prominent role in this selective sensitivity, crossing the blood–brain barrier and activating immune cascades [34]. Recent data suggest taxonomic alterations in gut microbiota with a decrease in commensal organisms can play a crucial role in multiple sclerosis via the modulation of the immune system [35]. 

Interestingly, factors associated with the alteration of microbial diversity among multiple sclerosis patients include dietary changes [35]. In multiple sclerosis patients, a bacterium called *Ruminococcus torques*, which is found in the gut, has been found to have a negative correlation with sodium intake [35]. Recent studies suggest that *Faecalibacterium prausnitzii*, a bacterium found in the intestines of MS patients, may have a potential role in the treatment of MS because of its anti-inflammatory and gut-promoting properties [35]. Interestingly, this bacterium has a positive correlation with fruit intake [35]. However, despite ample fruit intake, its levels remain low in MS patients [35]. These findings suggest that the gut microbiota of MS patients is affected by a complex disease-related pathology. Studies have also established possible communication between gut microbiota and T-cell chemokine receptor 9 and its ligand [36]. In Kadowaki et al.’s study, the decreased function of chemokine receptor 9 was seen in multiple sclerosis patients [36]. An experiment where germ-free mice were given antibiotics was conducted with a subsequent increase in chemokine receptor 9 [36]. This implied a relationship between microbiome manipulation and T-cell receptors [36]. To further consolidate the results, mice were induced with autoimmune encephalitis followed by a short course of antibiotics [36]. Antibiotics led to a significant improvement in autoimmune encephalitis along with an increase in T-cell receptors, further consolidating the relationship [36].

Studies have demonstrated that individuals suffering from multiple sclerosis exhibit a reduction in the amount of *Prevotella* and *Parabacteroides* bacteria in their fecal matter [37]. These microorganisms are responsible for mitigating inflammation in mice, thereby indicating a potential link between gut bacteria and the development of multiple sclerosis [37]. Conversely, a bacterium that is responsible for the differentiation of T-cells is found in abundance among multiple sclerosis patients, that is, *Akkermansia muciniphila* [37]. Other toxic metabolites of the gut microbiome include trimethylamine-N-oxide (TMAO), a vascular toxin noted to have a role in multiple sclerosis disease pathology [38]. On the other hand, secondary bile acids produced by our gut microbiome have a neuroprotective role in modulating glial and myeloid cell activation within the central nervous system [39]. Vitamin K also has been found to have an emerging role in neurodegenerative diseases, including multiple sclerosis [40]. Vitamin K, along with its role as an antioxidant, has a concentration-dependent role in the activation of the TAM (Tyro3, Axl, and Mertk) family of tyrosine kinase receptors, which play a pivotal role in reducing the expression of proinflammatory molecules and preventing auto-immunity because of its potential role in myelination [40]. Significantly lower Vitamin K levels in patients with multiple sclerosis, as compared with the general population, have been noted and may also be explained by low production given alterations in the host gut microbiome [40]. 

After treatment with fecal microbiota transplant, it was noted that multiple sclerosis patients had an increase in *Pseudomonas*, *Blautia*, *Streptococcus*, *Akkermansia*, *Ruthenibacterium lactatiformans*, *Hungatella hathewayi*, and *Eisenbergiella tayi* [35]. Notable decreases in *Prevotella*, *Bacteroides*, *Parabacteroides*, and *Clostridia* species were seen with fecal microbiota transplant; these organisms produce compounds that increase gastrointestinal permeability, exposing our bodies to new antigens and triggering an autoimmune response [29]. Furthermore, *Adlercreutzia* was found to be the most prominent genus in mice with autoimmune encephalitis treated with fecal microbial transplant [41]. *Adlercreutzia* has a negative correlation with inflammatory genes [41]. This genus modulates differentially expressed genes in the spinal cord, altering immune response after fecal microbial transplant [41]. Another change that was noticed in mice treated with fecal microbial transplant was the suppression of anti-inflammatory cytokines, including interleukin-10, thus suppressing the anti-inflammatory response [42].

Our reviewed studies revealed increased gastrointestinal permeability in 20–73% of the multiple sclerosis patients, suggesting modulation in the gut microbiome triggering more numerous immune responses to antigens [29]. The gut microbiome also modulates disease pathology via metabolites (pathologic versus protective), with major end products from anaerobic bacteria being short-chain fatty acids (SCFAs) including propionate, acetate, and butyrate [43,44,45,46,47]. SCFAs can cross the blood–brain barrier through passive and active transport, influencing neurotransmitter production, mitochondrial function, immune activation, lipid metabolism, and gene expression. Above all, the accumulation of SCFAs can acidify the pH, intracellularly modifying calcium signaling and gap junction inhibitions and ultimately affecting neuronal communications and behavior [43,44,45,46,47]. As highlighted in Engen’s study, butyrate-producing bacteria assist in regulating intestinal permeability and immune system responses, ultimately leading to a notable improvement in multiple sclerosis patients [30]. By introducing fecal microbiota transplant from donors not affected by multiple sclerosis, it stands to reason that more butyrate-producing bacteria are also be introduced, assisting in replenishing previously depleted SCFAs [43,44,45,46,47]. Notably, butyrate-producing organisms have been found to assist in increasing low levels of brain-derived neurotrophic factor (BDNF), which plays a crucial role in the function and development of neurons [43,44,45,46,47].

Our secondary goal was to demonstrate the safety of fecal microbiota transplants in multiple sclerosis patients. Overall side effects and the need for hospitalization determine the overall safety of fecal microbiota transplant. Common side effects associated with fecal microbiota transplant include gastrointestinal symptoms like diarrhea, boating, discomfort, flatulence, vomiting, and transient fever. In our reviewed studies, only one patient experienced an adverse reaction related to fecal microbiota transplant, which was hives during the initial treatment. This resolved on its own without intervention and did not recur in repeated fecal microbiota transplant infusions [29]. While data on the use of fecal microbiota transplant for neuromodulation are newer, there is concern for the transfer of occult infection via donor stool, requiring long-term follow-up. The safety of fecal microbiota transplant in multiple sclerosis patients is unexplored, with minimal data currency available. Despite its efficacy, there remain concerns regarding its potential to transmit infectious organisms, thus worsening multiple sclerosis symptoms and acute respiratory distress syndrome [48]. Despite its efficacy, important barriers are the rigorous screening required before the procedures and ethical dilemmas, increasing the cost and duration of the process. Rigorous screening can counter the transmission of infections with the development of innovative strategies to overcome technical difficulties. Dedicated facilities with specialists, innovative screening methods with online surveys, and increased awareness among the general population can somewhat counter these difficulties. 

## 5. Limitations and Conclusions

The primary limitation is the small number of studies and recruited patients in each study utilizing fecal microbiota transplant in multiple sclerosis. We also included three case studies and a case series with fewer than five patients, which had low-quality evidence, but we decided to include all studies because of the paucity of data on the subject. Each study had different fecal microbiota transplant administration protocols, such as different bowel preparations and pretreatments with antibiotics before fecal microbiota transplant, which could be a confounder. Given the limited data, patients were not sub-classified with different types of multiple sclerosis during assessment. The patients’ baseline diets, treatment/medications, and environments were also not accounted for during the follow-up period; while newer data suggest that these factors should not have a significant impact on gut microbiota, there may be some alteration in the ability of the transplanted organisms to effectively colonize their area of implantation. 

Despite these limitations, our review does suggest an improvement in the motor symptoms of multiple sclerosis patients undergoing fecal microbiota transplant, which stood the test of time, with the longest follow-up showing remission even after 15 years with no side effects. These patients also underwent repeated fecal microbiota transplant infusions without any significant adverse reactions, highlighting the safety of fecal microbiota transplant in multiple sclerosis patients. Notwithstanding these findings, more multi-center randomized clinical trials are required to further assess and consolidate these results for the widespread implementation of this novel technique, which has the potential to prevent the debilitating effects of multiple sclerosis. 

## Figures and Tables

**Table 1 microorganisms-11-02840-t001:** Types of multiple sclerosis [18].

Relapsing–Remitting Multiple Sclerosis (RRMS)	Episodes of acute exacerbations followed by recovery with an intermittently stable course
Secondary Progressive Multiple Sclerosis (SPMS)	Neurologic deterioration gradually with worsening symptoms (with or without relapses) in an RRMS patient
Primary Progressive Multiple Sclerosis (PPMS)	Continuous gradual neurologic deterioration (no relapses or remissions)
Progressive-Relapsing (PRMS)	Gradual neurologic deterioration (with subsequent relapses but no remissions)

**Table 2 microorganisms-11-02840-t002:** A summary of the included studies.

Author/Year	Study Design	Location (Country)	Treatment Before Fecal Microbiota Transplant	Intervention	Follow-Up
Kait 2022 [29]	Randomized controlled trial	Canada	None	Fecal microbiota transplant	Once per month for up to 1 year
Engen 2020 [30]	Single-arm, non-randomized, time series	Chicago	None	Fecal microbiota transplant	3, 13, 26, 39 weeks and 1 year follow-up
Makkawi 2017 [31]	Case study	Houston	Glatiramer acetate	Fecal microbiota transplant	Periodic follow-up over 10 years
Thomas Borody, 2011 [32]	Case series	Australia	Mexiletine, tryptanol, and B-interferon	5–10 fecal microbiota transplant infusions	8 months; 2, 3, 14 years
Victor Garcia-Rodriguez 2020 [33]	Case study	Canada	Vancomycin, metronidazole, fidoxamicin	Lymphosized fecal microbiota transplant orally	1 week and 1 year

**Table 3 microorganisms-11-02840-t003:** Summary of patient population and their baseline symptoms before fecal microbiota transplant.

Author/Year	Population Characteristics	Sample Size (n)	Mean/Median Age	Male/Female (n)	Neurologic Scoring Before Fecal Microbiota Transplant	Neurological Symptoms Before Fecal Microbiota Transplant
Kait 2022 [29]	Baseline EDSS score, 3.0 (n = 9)	9	**40.3 ± 11.7 years**	3/6	EDSS	Not reported
Engen 2020 [30]	Adult multiple sclerosis patient with bloating	1	42	1/0	MSWS-12	Abnormal gait
Makkawi 2017 [31]	Adult multiple sclerosis patient with recurrent Clostridium difficile infection	1	61	0/1	MSFC and EDSS	Worsening balance, ambulation, lower limb power, bladder function, and fatigue
Thomas Borody, 2011 [32]	Adult multiple sclerosis patients with constipation	3	**30 (median)**	2/1	None	Severe leg weakness resulting in difficulty walking
Victor Garcia-Rodriguez 2020 [33]	Adult multiple sclerosis patient with recurrent Clostridium difficile infection	1	52	0	EDSS	Horizontal nystagmus, ⅖ muscle strength in the right arm and leg, and increased deep tendon reflexes bilaterally

MSFC: Modified Multiple Sclerosis Functional Composite, EDSS: Expanded Disability Status Scale.

**Table 4 microorganisms-11-02840-t004:** A summary of Kait’s gastrointestinal and neurological symptoms [29].

Neurological Scoring Before Intervention	Intervention	Neurological Symptoms Post-Fecal Microbiota Transplant Infusions	Neurological Imaging Post-Fecal Microbiota Transplant Infusions
Baseline EDSS score, 3.0 (n = 9)	Fecal microbiota transplants every month for six months (6 patients); 3 received at least one.	Baseline EDSS score, 3.0 (n = 9). EDSS score was measured at every visit with no significant change in EDSS following repeat fecal microbiota transplants.	MRI at baseline and following fecal microbiota transplants did not show any new lesions.

EDSS: Expanded Disability Status Scale.

**Table 5 microorganisms-11-02840-t005:** A summary of Engen’s neurological and gastrointestinal symptoms [30].

Neurological Symptoms Before Intervention	Intervention	Gastrointestinal Symptoms After Intervention	Neurological Symptoms After Intervention
Significant tingling sensations in the extremities with abnormal gait	Fecal Michael round transplant	Resolved	Enhanced walking and balance metrics after fecal microbiota transplant with improvement in 5/6 metrics at 52 weeks as compared with baseline.

**Table 6 microorganisms-11-02840-t006:** A summary of Makkawi’s neurological and gastrointestinal symptoms at follow-up post-fecal microbiota transplant [31].

Neurological Symptoms Before Intervention	Intervention	Gastrointestinal Symptoms After Fecal Microbiota Transplant	Neurological Symptoms after Fecal Microbiota Transplant
Worsening balance, ambulation, lower limb power, bladder function, and fatigue	Fecal microbiota transplant for recurrent Clostridium difficile infection	Not reported	Stabilization of progression of multiple sclerosis with slight improvement at 10-year follow-up

**Table 7 microorganisms-11-02840-t007:** A summary of Borody’s neurological and gastrointestinal symptoms at follow-up post-fecal microbiota transplant [32].

	Neurological Symptoms with Scoring Before Intervention	Intervention	Gastrointestinal Symptoms After Intervention	Neurological Symptoms After Intervention
30-year-old male with multiple sclerosis	Severe leg weakness requiring a wheelchair and an indwelling catheter	5 fecal microbiota transplant infusions for constipation	Resolved	Post-fecal microbiota transplant patient’s multiple sclerosis progressively improved, restoring his ability to walk and facilitating the removal of his catheter. Patient remained relapse-free for 15 years post-fecal microbiota transplant.
29-year-old male with multiple sclerosis	A wheelchair-bound male with paresthesia and leg muscle weakness	10 days of fecal microbiota transplant infusions for chronic constipation	Resolved	Progressive improvement in neurological symptoms; regained ability to walk following slow resolution of leg paresthesia. Normal motor, GI, and urinary function at 3-year follow-up.
80-year-old female with multiple sclerosis	Severe muscular weakness resulting in difficulty walking	5 fecal microbiota transplant infusions for severe chronic constipation	Resolved at 8-month follow-up	Unassisted walking for long distances at 8 months and asymptomatic 2 years post-fecal microbiota transplant

GI: gastrointestinal.

**Table 8 microorganisms-11-02840-t008:** A summary of Rodriguez’s neurological and gastrointestinal symptoms at 1 week and 1 year follow-up [33].

Neurological Symptoms with Scoring Before Intervention	Intervention	Gastrointestinal Symptoms at 1 Week	Neurological Symptoms at 1 Week	Neurological Symptoms at 1 Year
Horizontal Nystagmus, ⅖ muscle strength in the right arm and leg, and increased deep tendon reflexes bilaterally; EDSS, 8.5	Lymphosized fecal microbiota transplant orally using a standard protocol for recurrent Clostridium difficile infection	Completely resolved	No change	Improvement of right upper extremity strength with a slight decrease in EDSS score to 8

EDSS: Expanded Disability Status Scale.

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
