# Peer review of "The Potential Role of Fecal Microbiota Transplant in the Reversal or Stabilization of Multiple Sclerosis Symptoms: A Literature Review on Efficacy and Safety"

_microorganisms, 2023, doi:10.3390/microorganisms11122840_

Round 1

Reviewer 1 Report

Comments and Suggestions for Authors

Dear Authors,

 The submitted review is within the Journals scope and shows the potential to attract the readerships attention. During the review, the concerns appeared that necessitated a detailed revision before the acceptance for publication. Please find below the detailed comments and suggestions.

 Title, abstract, and keywords

 - Scoping does not seem necessary in the Title and throughout the Manuscript.

- According to the Instructions to Authors, the Abstract should not contain more than 200 words.

- Lines 24: Please consider removing “first of its kind scoping”. The suggestion also refers to the text (Line 109).

- Line 25: The details about search engines would be appreciated.

- Line 26: Please clarify whether 150 patients were included in the total or per study.

- Lines 30–1: Please emphasize that the mentioned patients were from the published studies.

- Line 33: The follow-up duration would be appreciated.

- Keywords are missing.

 Introduction

- Lines 41–3: Do the percentages indicate the survey response rate or the global frequency of countries with reported multiple sclerosis (MS) prevalence?

- Line 50: The term evidence of cytomegalovirus in the serum seems elusive. Please provide an additional explanation. The analogous comments are valid for temporary regions (Line 55) and plasma therapy (Line 91).

- Lines 60–3: Please avoid repetition. The sentences bring almost identical info.

- Lines 66–72: These sentences might be a new paragraph. Please consider the additional efforts to include the main features from each diagnostic group. What are the most relevant issues encountered in practice?

 Methods

- Line 123: What was the minimum follow-up period?

- Lines 125–6: The second and third criteria were described in the search strategy.

- Line 149: Initials are identical.

 Results

*Table 2*

- There is an inconsistency between the title and the data format in the 3rd column. 

- What represented the standard protocol mentioned in the last row in the fourth column?

- In the 6th column, please convert the weeks into the years, where appropriate.

*Table 3*

- Mean is not suitable for the title of the 4th column. For studies 1 and 4, please provide age as median with minimum and maximum.

- The 5th and 6th columns could be merged, with the number of males and females separated by a slash.

- Please consider merging and reorganizing Tables 4–8 to make them less challenging for follow-up.

- Table 9 does not seem necessary because the contained info could be easily described.

 Discussion

 - Line 194: Please consider the additional efforts to be more specific about the follow-up duration.

- Lines 204–9: Please indicate the body compartment(s) with the high prevalence.

- Lines 218–20: The sentence is challenging to interpret. Please consider rephrasing.

- Line 247: The reviewed studies seem more suitable than our included studies. An analogous suggestion is valid for our systemic review in Line 269.

- Line 298: Notwithstanding would be more suitable than despite.

 Technical suggestions

- Language editing would contribute to the overall quality of the Manuscript.

- Please consider additional efforts to explain all abbreviations and to properly use the Greek letters and the Latin titles.

- Please avoid using different fonts.

Comments on the Quality of English Language

The details are within the Comments and Suggestions for Authors.

Author Response

Title, abstract, and keywords

 - Scoping does not seem necessary in the Title and throughout the Manuscript.

- Deleted 

  • According to the Instructions to Authors, the Abstract should not contain more than 200 words.

- Fixed

  • Lines 24: Please consider removing “first of its kind scoping”. The suggestion also refers to the text (Line 109).

- Removed 

-Line 25: The details about search engines would be appreciated.

Added

  • Line 26: Please clarify whether 150 patients were included in the total or per study.

- clarified 

  • Lines 30–1: Please emphasize that the mentioned patients were from the published studies.

- Emphasized 

  • Line 33: The follow-up duration would be appreciated.

- Duration added

  • Keywords are missing.

They were submitted at the time of submission. If not found, can submit it again. 

 Introduction

  • Lines 41–3: Do the percentages indicate the survey response rate or the global frequency of countries with reported multiple sclerosis (MS) prevalence?

- It represents the countries who responded 

  • Line 50: The term evidence of cytomegalovirus in the serum seems elusive. Please provide an additional explanation. The analogous comments are valid for temporary regions (Line 55) and plasma therapy (Line 91).

Additional details added 

  • Lines 60–3: Please avoid repetition. The sentences bring almost identical info.

Sentence edited 

  • Lines 66–72: These sentences might be a new paragraph. Please consider the additional efforts to include the main features from each diagnostic group. What are the most relevant issues encountered in practice?

- Main features added in Table 1. Please let me know if more detail is needed. 

 Methods

  • Line 123: What was the minimum follow-up period?

Minimum follow-up period added

- Lines 125–6: The second and third criteria were described in the search strategy.

  • Line 149: Initials are identical.

 Results

*Table 2*

  • There is an inconsistency between the title and the data format in the 3rd column.

Unclear - title edited 

  • What represented the standard protocol mentioned in the last row in the fourth column?

- Standard protocol criteria not mentioned in the study 

  • In the 6th column, please convert the weeks into the years, where appropriate.

- years added 

*Table 3*

  • Mean is not suitable for the title of the 4th column. For studies 1 and 4, please provide age as median with minimum and maximum.

- for study 1, the median is not described in the study. For study 4 median added

- The 5th and 6th columns could be merged, with the number of males and females separated by a slash.

  • Please consider merging and reorganizing Tables 4–8 to make them less challenging for follow-up.

- Given the different end points described in these studies, it might be difficult to make one table out of them 

  • Table 9 does not seem necessary because the information contained could be easily described.

Already described in the end of the discussion. Can delete the table? 

 Discussion

 - Line 194: Please consider the additional efforts to be more specific about the follow-up duration.

- Added

  • Lines 204–9: Please indicate the body compartment(s) with the high prevalence.

- Added

  • Lines 218–20: The sentence is challenging to interpret. Please consider rephrasing.

- Rephrased

  • Line 247: The reviewed studies seem more suitable than our included studies. An analogous suggestion is valid for our systemic review in Line 269.

- Changed 

  • Line 298: Notwithstanding would be more suitable than despite.

- Changed 

Reviewer 2 Report

Comments and Suggestions for Authors

1. It is suggested to appropriately delete part of the introduction and simplify the diagnosis and treatment of multiple sclerosis.

2. Patients with other diseases shall be considered in the exclusion criteria.

3. Most of the content in the discussion originates from the same literature, and the discussion lacks depth and pertinence.

4. Discuss inconsistent fonts in multiple paragraphs.

Comments on the Quality of English Language

None

Author Response

  1. It is suggested to appropriately delete part of the introduction and simplify the diagnosis and treatment of multiple sclerosis.

- Deleted 

2. Patients with other diseases shall be considered in the exclusion criteria.

- Added

3. Most of the content in the discussion originates from the same literature, and the discussion lacks depth and pertinence.

-Edited 

4. Discuss inconsistent fonts in multiple paragraphs.

-Fixed 

Reviewer 3 Report

Comments and Suggestions for Authors

This review article deals with an interesting topic related to multiple sclerosis and its relationship with gut microbiota transplants. It has a good quality and excellent level of scientific rigor. However, minor changes are needed to be publishable in microorganisms journal.

Detailed comments and suggestions are in the file I've attached, so please revise them; I will be honored to review the corrected version of this manuscript.

Comments on the Quality of English Language

Minor editing of English language required

Author Response

- Abstract shortened

- Web of Science and Scopus were also searched however no studies were found and included from those search engines

- Antibiotics were not used in the exclusion criteria because of lack of data to control for it as most places use pre-procedural antibiotics

- Typos corrected

- No genus for Vitamin K identified in the study

- genus names italicized

- Reference added

Round 2

Reviewer 1 Report

Comments and Suggestions for Authors

Dear Authors,

Your efforts to improve the overall quality of the Manuscript are evident and appreciated. However, the issues remained and necessitated another review round. Please find below the comments and suggestions.

Abstract

- The word count still exceeds 200, the maximum allowed by the Instruction for Authors.

- Line 20: Please rephrase to avoid ambiguity. The term screening criteria is elusive. Furthermore, the sentence implies that you removed those criteria.

- Lines 26 and 32: Please avoid referencing that the patients were included in your study. The review comprehensively summarized the published data but did not bring results from the research conducted on your own. The comment refers also to the text of the Manuscript.

- Keywords are lacking.

Introduction

- Lines 49–52: What is the meaning of the term evidence of cytomegalovirus (CMV) in the serum? Please explain whether you referred to the presence of the viral particles or the antibodies. Furthermore, associating CMV presence in serum and multiple sclerosis (MS), a neurologic pathology, is challenging and merits a more cautious explanation.

- Lines 61–3: The sentence seems more suitable for the beginning of this paragraph.

- Lines 67–73: What are the main MRI and lab findings indicative of MS? Which diagnostic challenges remained after introducing the revised criteria in 2017?

Material and Methods

- Line 156: The initials are identical.

*Table 2*

- Treatment before fecal microbiota transplantation might be a more suitable title for the third column.

- For studies 3–5, please indicate the frequency of check-ups, analogously with studies 1 and 2.

Results

*Table 3*

- The 5th and 6th columns could be merged, with the number of males and females separated by a slash.

- Table 9 seems unnecessary because lines 176–80 contain the same information.

Discussion

- Line 237: In ParaBacteroides, only the first letter should be capitalized.

Author Response

- The word count still exceeds 200, the maximum allowed by the Instruction for Authors.

- Line 20: Please rephrase to avoid ambiguity. The term screening criteria is elusive. Furthermore, the sentence implies that you removed those criteria.

-Lines 26 and 32: Please avoid referencing that the patients were included in your study. The review comprehensively summarized the published data but did not bring results from the research conducted on your own. The comment refers also to the text of the Manuscript.

Thank you for your feedback regarding the abstract. It is re-written. Please give it a read and let me know if further editing is needed.

- Keywords are lacking. Added

Introduction

Lines 49–52: What is the meaning of the term evidence of cytomegalovirus (CMV) in the serum? Please explain whether you referred to the presence of the viral particles or the antibodies. Furthermore, associating CMV presence in serum and multiple sclerosis (MS), a neurologic pathology, is challenging and merits a more cautious explanation.

It indicates seropositivity for infection. Added in the manuscript. If argument seems weak, can remove CMV from protective factors altogether.

Lines 61–3: The sentence seems more suitable for the beginning of this paragraph.

Changed 

Lines 67–73: What are the main MRI and lab findings indicative of MS? Which diagnostic challenges remained after introducing the revised criteria in 2017?

Elucidated

Material and Methods

Line 156: The initials are identical.

The first and second authors have identical initials. Any recommendations for not having similar initials? There are no middle names either for the authors. 

*Table 2*

Treatment before fecal microbiota transplantation might be a more suitable title for the third column.

Changed 

For studies 3–5, please indicate the frequency of check-ups, analogously with studies 1 and 2.

For studies 3-5, follow-up periods are not as well defined as 1-2. However, I added more details that were available. 

Results

*Table 3*

The 5th and 6th columns could be merged, with the number of males and females separated by a slash.

Merged

Table 9 seems unnecessary because lines 176–80 contain the same information.

Deleted 

Discussion

Line 237: In ParaBacteroides, only the first letter should be capitalized.

Fixed